# OpenReview forum: "Investigating the Effects of Emotional Stimuli Type and Intensity on Large Language Model (LLM) Behavior"
_ICLR.cc/2025/Workshop/BuildingTrust — Submitted to BuildingTrust_

### Official Review · Reviewer_ibwQ · 2025-02-26
**Review of Investigating the Effects of Emotional Stimuli Type and Intensity on Large Language Model (LLM) Behavior**

**Rating:** 4
**Confidence:** 3

**Review:**

This research addresses an interesting and understudied dimension of prompt engineering - how emotional content affects LLM responses across multiple dimensions of performance. Furthermore, the creation of a "Gold Dataset" containing prompts that humans and LLMs agree on regarding emotion type and intensity is a valuable contribution. While the core premise of studying emotional prompting effects on LLMs is interesting and valuable, the implementation has several limitations that reduce the paper's impact.

The most concerning aspect is the methodology for measuring sycophancy. The authors define sycophancy through a "positivity score" where the model (GPT-4o mini) is asked to compare responses and determine which is more positive. This approach is fundamentally problematic for several reasons. The mean positivity score is particularly suspect, as it is derived by prompting the same model (GPT-4o mini) to make a binary choice between responses. Since the same or a similar model is used both to generate and to evaluate responses, this method may be inherently circular. It is not clear that GPT's own internal heuristics for "positivity" align well with the broader notion of sycophancy or with human perceptions of tone. This self-evaluation introduces significant bias into the results.

The accuracy results show remarkably small differences between conditions. For example, the percentage differences in Table 1 are consistently under 2% (with many under 1%), with joy and encouragement showing improvements of only 0.84% and 0.56% for LLM-generated prompts. Similarly, the toxicity results in Table 3 show differences ranging from -0.10% to -1.89%. These tiny effects raise serious questions about whether these differences are statistically significant or practically meaningful for real-world applications.

The experimental design raises additional concerns. The paper switches between one-shot and few-shot prompting in different parts of the methodology without clearly justifying these choices. For instance, zero-shot prompting is used for emotion detection while few-shot prompting is used to generate emotional prompts. The paper does not explain why these different approaches were chosen or how they might affect the results. Furthermore, the paper lacks essential statistical analysis - there are no error bars on the figures, no variance measures for the reported means, no p-values to indicate statistical significance, and no confidence intervals to understand the reliability of the findings. Without these statistical indicators, it's impossible to determine if the small observed differences are meaningful or simply noise in the data.

---

### Official Review · Reviewer_hxWy · 2025-03-01
**Investigating the Effects of Emotional Stimuli Type and Intensity on Large Language Model (LLM) Behavior**

**Rating:** 5
**Confidence:** 3

**Review:**

The paper investigates the influence of emotional prompts, categorized into positive (joy, encouragement) and negative (anger, insecurity) emotions, on large language model (LLM) behaviors in terms of factual accuracy, sycophancy, and toxicity.

Strengthen:
- The authors demonstrate methodological rigor by creating a gold dataset, ensuring human and model agreement on emotional labeling and intensity.

Weaknesses:
- The overall approach appears simplistic, primarily focusing on only four basic emotional categories. The nuanced complexity of emotional responses and their influence on language generation is insufficiently explored.
- Observed improvements in accuracy are minimal (typically less than 2%), raising doubts about the practical significance and utility of emotional prompts in real-world scenarios.
- Experiments are limited to a single LLM (GPT-4o mini), lacking comparative analysis with other models to assess generalizability.
- The operationalization of sycophancy via positivity scores might be overly simplistic

Suggestions:
- Expand the range of emotions studied, including more nuanced emotional categories to deepen the analysis and findings.
- Validate findings with multiple LLM architectures to strengthen claims of generalizability.

While the topic is relevant and interesting, the current paper's simplistic treatment of emotional stimuli and limited scope diminish its potential impact.

---

### Decision · Program_Chairs · 2025-03-05

**Decision:**

Reject

**Comment:**

This paper examines how emotional prompts (e.g., joy, encouragement, anger, insecurity) influence LLM factual accuracy, sycophancy, and toxicity. However, the observed effects are small (as noted by R1), and the study lacks relevance to the workshop’s focus. Given these limitations, I recommend rejection.